# Reinforcement Learning-Aided Edge Intelligence Framework for Delay-Sensitive Industrial Applications

**DOI:** 10.3390/s22208001

**Published:** 2022-10-20

**Authors:** Muhammad Zubair Islam, Rashid Ali, Amir Haider, Hyung Seok Kim

**Affiliations:** 1School of Intelligent Mechatronics Engineering, Sejong University, Seoul 05006, Korea; 2Department of Information and Communication Technologies, Universitat Pompeu Fabra, 08018 Barcelona, Spain

**Keywords:** 5G, URLLC, tactile Internet, IoT, codecs, RL

## Abstract

With the advancement in next-generation communication technologies, the so-called Tactile Internet is getting more attention due to its smart applications, such as haptic-enabled teleoperation systems. The stringent requirements such as delay, jitter, and packet loss of these delay-sensitive and loss-intolerant applications make it more challenging to ensure the Quality of Service (QoS) and Quality of Experience (QoE). In this regard, different haptic codec and control schemes were proposed for QoS and QoE provisioning in the Tactile Internet. However, they maximize the QoE while degrading the system’s stability under varying delays and high packet rates. In this paper, we present a reinforcement learning-based Intelligent Tactile Edge (ITE) framework to ensure both transparency and stability of teleoperation systems with high packet rates and variable time delay communication networks. The proposed ITE first estimates the network challenges, including communication delay, jitter, and packet loss, and then utilizes a Q-learning algorithm to select the optimal haptic codec scheme to reduce network load. The proposed framework aims to explore the optimal relationship between QoS and QoE parameters and make the tradeoff between stability and transparency during teleoperations. The simulation result indicates that the proposed strategy chooses the optimal scheme under different network impairments corresponding to the congestion level in the communication network while improving the QoS and maximizing the QoE. The end-to-end performance of throughput (1.5 Mbps) and average RTT (70 ms) during haptic communication is achieved with a learning rate and discounted factor value of 0.5 and 0.8, respectively. The results indicate that the communication system can successfully achieve the QoS and QoE requirements by employing the proposed ITE framework.

## 1. Introduction

The dramatic growth in communication technologies, specifically Fifth-Generation (5G), has obtained enormous interest due to its evolving generic services like massive machine-type communication, Ultrareliable and Low Latency Communication (URLLC), and enhanced mobile broadband [1,2,3]. Apart from this, it is one of the robust key enablers to realizing the so-called Tactile Internet, which provides to steer and control physical and virtual objects at a distance in real-time. The Tactile Internet is envisioned to enable human-to-machine communication where a human being can communicate with machines in a physical/virtual environment and experience haptic sensations (touch and forces) along with conventional audio–video traffic [4]. However, the strict requirements of Tactile Internet such as ultralow latency, ultrahigh reliability, high availability, and ultrasecurity remain open problems to enabling haptic communication over 5G. As per report Release 15 of the Third-Generation Partnership Project (3GPP), the URLLC service provides the reliability of 99.9% for a data packet of size 32-byte with communication latency of 1 ms [5]. Tactile Internet demands a higher haptic data packet rate (1 kHz) and reliability greater than 99.9% under 1 ms latency. Therefore, the system design and architectural limitations of 5G technologies are not promising to realize haptic communication within 1 ms latency [6]. The sixth-generation (6G) systems with learnable network architecture address these challenges. In the literature, a number of studies present open systems interconnection stack layerwise analysis, identifying the requirements and challenges in each layer corresponding to next-generation emerging applications [6,7]. The work in [8] explored the use cases and technical requirements of 6G systems. Similarly, the authors of [9] discussed the trade-off between computation, compression, bandwidth, and latency of the communication system, and the authors of [10] presented the Reinforcement-Learning (RL)-based resource allocation approaches for beyond 5G systems. The authors of [11] explored the zero-touch and network service management methods with a focus to enable the network to perform self-configuration and optimization based on application requirements. The authors present the network design, standard, and security aspects in the domain of zero-touch. The study in [12] revived the recent work on network security, privacy, and trust to realize the ultrasecure 6G communication network for emerging applications of 6G. In this work, we also present an overview of the 6G application along with security requirements.

In order to achieve 1 ms latency, Software-Defined Networking (SDN) and Mobile Edge Computing (MEC) are utilized in the core network of 5G and beyond networks [13]. SDN provides the facility to separate the control and data plane in the communication network. The control plan is responsible for decision-making for network traffic while the data plan has a response to forwarding the traffic. The authors of the work [13] proposed an intelligent core network for smart applications of Tactile Internet. The authors utilized the SDN and MEC in a multitier way to the E2E latency requirements of Tactile Internet. The proposed model consists of three layers: a hardware resource layer, a software layer, and an application layer. The application layer link with the software layer via application programming interfaces. The hardware layer provides the MEC infrastructure in a multilayer design to connect haptic devices. The last layer connects the remote application to interact with the operator. Similarly, the work in [14] explored technologies such as SDN and network function virtualizations (NFV) to realize Tactile Internet over mobile networks. The authors claim that integrating the SDN, NFV, and MEC is a promising solution to develop a general network infrastructure and support the Tactile Internet application.

Various standard organizations, for instance, the International Telecommunication Union, 3GPP, and the Institute of Electrical and Electronics Engineering (IEEE) have been operating to empower the existing and design new architecture to incorporate haptic traffic over a network. The Tactile Internet IEEE P1819.1 standard group has been introduced to describe the basic definition of the Tactile Internet and design for standard communication architecture along with technical functions [15]. In addition, it defined Tactile Internet use cases including automotive, cooperative automated driving, Internet of drones, immersive virtual reality, haptic-enabled interpersonal communication, live haptic broadcast, and teleoperation with corresponding requirements.

The scope of this paper is to ensure the Quality of Services (QoS) and Quality of Experience (QoE) requirements for teleoperation systems with haptic feedback. The reason behind considering the teleoperation systems only is that most haptic-driven applications utilized teleoperations to interact with the object at a remote location. The basic architecture, along with key components of the haptic-enabled teleoperation system, is illustrated in Figure 1. The teleoperation system comprises three main components: the master domain, the network domain, and the slave domain. A human operator at the master side uses a haptic device and control commands are forwarded to the slave domain over the communication network. The network domain attaches the master and slave domains to provide bidirectional communication over the network. Teleoperators or controller robots are included in the slave domain and are directly controlled by the master domain via control signals. The master domain receives feedback from the slave domain, including haptic and audio–visual signals. Contrary to the traditional communication technological services, Tactile Internet teleoperation service demand includes haptic data packet rate (1 kHz or higher), the packet loss rate $10^{-3}$–$10^{-7}$ availability (99.99%–99.99999%), and End-to-End (E2E) delay in milliseconds (1 ms) [16,17]. Moreover, these requirements rely upon the application’s nature. For example, mission-critical applications, such as telesurgery, required the reliability of around $10^{-7}$ with a latency of 1 ms, while teletherapy for phobia treatment demands E2E latency of around 50 ms and remote industrial management requires between 0.25–10 ms. The main challenge for Tactile Internet realization is the transmission of haptic traffic with a packet rate ≥1 kHz under low latency. The high packet rate induces network congestion that leads to packet delay and large packet loss and makes the system prone to QoS violation.

These stringent requirements expose different challenges to guaranteeing the QoS and QoE provisioning for the emerging services of Tactile Internet. In the literature, various solutions have been proposed based on haptic codecs and control approaches to tackle the above-mentioned teleoperation challenges [18]. The recent studies focus on integrating different haptic data compression/reduction approaches with stability-ensured control methods, as presented in [18,19]. However, the aforementioned studies achieve stability of the system during interaction while degrading the quality of interaction. Figure 2 shows the dynamic movement of the human operator’s hand by interacting with the moveable cube in the virtual application, and the dataset is adopted from the reference database of Tactile Internet haptic codec working group (IEEE P1918.1.1) [19]. Furthermore, it can be clearly observed from Figure 2 that most of the time, environment (operator/teleoperator) dynamics remain away from their peak values. As a result, the demands of the less dynamic environment for latency, reliability, availability, and security can be eased. The high level of environmental dynamicity involves stringent requirements to guarantee QoS and QoE. If we allocate the required QoS statically as per the demand of the application, then we get the guarantee for the desired QoE. However, the allocated network resources are unexploited most of the time.

To this end, there is a strong need for a strategy to achieve the tradeoff between stability and transparency of the system and meet the QoS and QoE requirements under variable time delay and high packet rate. Therefore, in this paper, an RL-based dynamic switching strategy has been proposed where the haptic codecs govern the allocated environment’s (operator/teleoperator) dynamics and network resources. The proposed strategy estimates the network resources in real-time and utilizes one of the famous RL algorithms, Q-learning, to select the optimal haptic codec schemes. The proposed switching strategy aims to gain both stability and transparency while meeting the stringent QoS and QoE requirements of tactile Internet services like teleoperations. The proposed strategy collects the Explicit Congestion Notification (ECN) marking packets during communication and inputs to the Long Short-Term Memory (LSTM) model to estimate the congestion levels in the network and select the best codec scheme to reduce the congestion and provide the desired QoS and QoE. The core network design of the proposed system comprises SDN, and MEC technology to realize reduce network latency and support Tactile Internet applications. As we discussed, the SDN paradigm reduces the network complexity and provides dynamic sharing of network resources. Moreover, it provides flexibility to the network, where the operators can define virtual slices corresponding to use cases with a focus to ensure QoS. The major contributions of this paper are summarized below as:We discuss the Tactile Internet challenges in the context of high pack rate/packet size and extensively review different haptic codec solutions and summarized their key contributions.An Intelligent Tactile Edge (ITE) framework is proposed that incorporates the ECN mechanism, Machine Learning (ML), and particularly, RL policies for optimal haptic codecs selection based on network congestion level employing the Markov decision process (MDP).In the proposed framework, an LSTM model is utilized to predict the congestion level in the network by exploiting ECN information in the TCP header of the data packet. To select the best haptic codec scheme corresponding to the network congestion, an RL-based Q-learning algorithm is employed.We conduct different experiments to demonstrate that the proposed system selects the best haptic data compression and reduction scheme for haptic-enabled bilateral teleoperations to ensure QoS and QoE requirements.

The remainder of this paper is structured as follows. Section 2 provides state-of-the-art research work on haptic codec schemes to provision QoS/QoE for haptic-driven teleoperation services. Section 3 presents the proposed strategy to predict network congestion and select optimal solutions to the corresponding QoS and QoE parameters. Section 4 demonstrates the experimental results in terms of QoS and QoE provisioning. Section 5 presents a discussion and future challenges. Finally, Section 6 concludes this paper and presets possible future directions.

## 2. Related Research Work

In the literature, various schemes have been suggested to address the communication challenges, such as the high packet rate of the haptic data and the time-varying delay of the network. The haptic data sampling rate challenge imposes a strong demand to packetize and transmit haptic information with a data rate 1 kHz or higher to provide stability and transparency to the teleoperation systems [20].

In [21], an approach termed as Opportunistic Adaptive Haptic Sampling (OAHS) was proposed to provide a higher data rate and signal transmission during haptic communication over the network. The Webers fraction was employed to guarantee the QoS requirement of Tactile-driven services. A data-driven technique to decrease the packet rate of haptic data in case of limited channel bandwidth was proposed in [22] with a focus to guarantee QoS and QoE of the communication system. The authors claim that their proposed scheme improves the application service quality while minimizing computational complexity and latency. The effectiveness of the data-driven was evaluated based on energy consumption. The recent work in [23] presents a Perception-Lossless Codec (PLC) approach to meet the E2E delay requirement of the haptic-derive applications of the Tactile Internet. It utilizes run-length coding to reduce the coding delay, and coding loss and improves the Rate-Distortion (RD) performance.

The work in [24] offers a Perceptual Vibrotactile Coding (PVC) derived from Sparse Linear Prediction (SLP) combining a cutaneous sensitivity model to assist haptic-driven applications. PVC-SLP scheme calibrates the filter parameter from the vibrotactile signal to produce signals. The performance of the PVC-SLP scheme was measured on two public datasets, with 280 and 1001 vibrotactile signals. It is considered a reference standard for tactile codecs. In [25] a Perceptual Wavelet Quantization (VC-PWQ) was proposed for vibrotactile codecs, where input signals are split into multiple blocks to feed the Wavelet which then inputs these blocks into a different frequency spectrum. Then perceptual thresholds were measured using a psychophysical model to lower the signal correlations. The presented method performed well in terms of SNR and PSNR across standard datasets with 280 vibrotactile samples using various compression ratios The authors in [26] explored the bursty haptic traffic problems of high packet rate during teleoperation that exposes several network issues like transmission delay and packet failure. A peak-suppressing adaptive Perceptual Deadband (PDb) approach was proposed which dynamically manages the packet rate based on earlier packet transmission traces. The proposed algorithm is capable of adjusting the PDb with a focus to minimize transmission delay.

Similarly, the study in [27] presents a perceptual-based adaptive sampling scheme to automatically predict the just noticeable difference and reduce the traffic rate in real-time. The proposed approach adjusts the Weber threshold without prior knowledge to maintain the QoE. However, their proposed solution only outperformed when compared against a predetermined set of QoS and QoE measures, such as a system with no communication delays that is also transparent. To maximize the QoE and improve the RD, a deep neural network (DNN)-based end-to-end vibrotactile autoencoder was presented in [28]. Table 1 provides a summary of recently proposed haptic data codecs to enhance QoS and QoE provisioning for teleoperation systems. The improved QoS/QoE factors column indicates which QoS and QoE metrics were improved by the reviewed study. This includes the different haptic quality measuring metrics such as Signal-to-Noise Ratio (SNR), Peak Signal-to-Noise Ratio (PSNR), Structural Similarity Index Measure (SSIM), Haptic Structural Similarity Index Measure (HSSIM), Mean Square Error (MSE), and Mean Opinion Score (MOS). The MOS is a subjective QoE evaluation metric while others are objective metrics. The PSNR measures the signal power by dividing the maximum power by the distorted signal power, whereas, the SNR is the ratio of signal power to signal noise power. The SSIM and HSSIM measure the structural similarity after data compression. The Effectiveness column explains the level of significance of the explored study. We categorized the significance into three levels, Significant (Sig.), Not Significant (Not Sig.), and Mixed. The detail of these levels is explained in the caption of Table 1.

In this paper, we proposed an ITE framework that dynamically selects the haptic codec scheme corresponding to the level of congestion in the network while attaining the desired QoE performance under variable network resources and required QoS.

## 3. Proposed Framework

The proposed strategy aims to predict the level of congestion in the network, compare them with the QoS and QoE requirements of the teleoperation system, and finally provide the optimal QoS and QoE solution to ensure the stringent requirements. To achieve this goal, the proposed RL-based ITE framework deals with the main two challenges: (1) estimate the network congestion and comparison with domain-specific requirements in real-time, and (2) select the optimal haptic codec to decrease the data rate without degradation in QoE performance and improve the system stability and transparency. This section presents a detailed discussion of the proposed ITE to deal with the aforementioned challenges. At first, we will discuss the ECN mechanism and how ECN-marked packets are used to predict the levels of congestion in communication networks. Secondly, we briefly discuss the Q-learning algorithm to select the optimal codecs based on the level of congestion.

### 3.1. Network Congestion Prediction

Figure 3 depicts the structure of the proposed ITE with a haptic-enabled bilateral teleoperation system scenario in line with the IEEE P1918.1 Tactile Internet reference architecture. The human operator at the master side uses the haptic device to control the remote objects at the salve domain over the communication network and received haptic feedback. The core network comprises SDN-based technology that enables the separation of the control and data plane and realizes Tactile applications. The SDN employs the OpenFlow protocol to communicate between the SDN switch and controller. The solid line indicated the transmission link weights in ms and the yellow dotted lines indicate the SDN control link with SDN-enabled switches. As we discussed in the previous section haptic-enabled services demand high QoS and QoE. As the packet size, and sampling rate for haptic traffic increase the network system induces congestion problems. To avoid congestion problems Active Queue Management (AQM) techniques are used that avoid the buffers overflow by dropping or marking the before buffer overflow [29,30]. In the literature, a lot of studies on AQM schemes have been suggested. The process of marking the data packets instead of dropping them when the network encounters incipient congestion is called ECN. The ECN process helps to reduce packet loss and latency. The data including ECN-marked packets help to understand the nature of the application and the level of congestion in the network. In our proposed ITE we utilized the ECN mechanism and take benefit of the Explicit Congestion Experienced (ECE) flag to collect the ECE-marked packets during haptic traffic transmission among the master and the slave. The ECE-marked packets are input into the LSTM model to forecast the level of congestion in the network. LSTM is a special form of recurrent neural network with memory blocks that make it efficient to learn long-term dependencies. The LSTM model comprises a cell, an input, an output, and a forget gate. LSTM mitigate the vanishing-gradient problem of recurrent neural network. In [31], the authors proposed intelligent active queue management using ECN, where they, utilized the neural network to estimate the congestion and improve the AQM algorithms based on predicted congestion. We borrowed the congestion prediction concept from the work present in [31] to predict the congestion on the rest of the path in real-time. The LSTM model was considered with three hidden layers and the following formula is used to calculate the number of neurons in each layer, it is defined as follows:(1)$$N_{n}=\left(N_{in}+\sqrt{N}\right)/L,$$
where the $N_{in}$ is the number of inputs to the LSTM layer, *N* refers to the number of samples and *L* is the number of hidden layers. Around 30 neurons are selected for each hidden layer in the LSTM model. The model is trained on 6000 samples at 100 ms intervals for ten minutes period. The proposed ITE deploys the trained model at the edge, which is also known as the tactile support engine in the IEEE P1918.1 Tactile Internet reference architecture. At each iteration, the model predicts the congestion level in the network and inputs the Q-learning algorithm to select the appropriate action to provide the required QoS/QoE. The work in [32] motivates us to integrate the advantages of the RL algorithm and LSTM-based prediction method in resource management. The proposed utilizes LSTM to predict user mobility and an actor-critic algorithm to allocate the network resources in serval slices of the network.

### 3.2. Q-Learning Technique

In this paper, we formulate the dynamic section of the QoS, and QoE provision approaches as the Markov decision process (MDP), where the ITE algorithm selects optimal policy as an agent and the communication network behaves as an environment. The intelligent agent communicate with the environment in sequential decision-making process to observe the current state s(t), perform an action a(t) at time *t*, and move it into the next state s(t+1). The state of the system, action, and reward function for the proposed system can be summarized as follows:*Agent*: An agent is an entity, which takes out learning tasks in the system. In the proposed framework the ITE that recommends the haptic codec at the application layer acts as an agent.*State*: A state of the understudy system provides the observation of the environment that can be examined by the agent. In our proposed framework, we define the discrete level of network congestion as the states S=s(1),s(2),s(3),…,s(T).*Action*: An action reveals how an intelligent agent responds to the environment based on the observed state. In ITE, the action *A* is the set of the haptic codecs schemes, which reduce and compress the packet size or packet rate of haptic traffic during communication to ensure QoS and QoE provisioning. The agent performs an action from the action space of databases of haptic codecs approaches and is denoted as $A=\left\{a_{0},a_{1},a_{2},\ldots ,a_{n}\right\}$, where *n* refers to the total available haptic codecs as listed in Table 1. In our case, we have a total of seven codecs, so n=7 and action space is $A=\left\{a_{0},a_{1},a_{2},\ldots ,a_{7}\right\}$.*Reward*: The reward to the system is a scalar value. At each time, when an agent executes an action, the environment returns a reward as a response to the agent. The reward characterizes the behavior (good/bad) of the environment to action and the agent adjusts its policies based on the reward. In this system, our aim is to increase the network performance and provide QoS/QoE by minimizing the congestion that leads to delay and packet loss. The ITE uses the power function of the connection as a reward function and defined it as follows:
(2)$$R(s_{t},a_{t})=Tput/RTT,$$
where the Tput is the throughput and Round Trip Time (RTT) total E2E delay that a packet experiences. So, the reward is explained as the ratio of throughput to RTT.

In the MDP, the successor state $s_{t+1}$ of the system merely relies on the current state $s_{t}$ of the system instead of the entire prior knowledge. Figure 4 depicts the overview of the optimal haptic codec scheme selection process of the proposed ITE based on the level of network congestion. As it can be seen from Figure 4, in this randomized procedure, when agents interact with the system during learning episodes, the records of the agent’s interaction with systems are recorded as a sequence of state, action, and rewards $\left\{s(1),a(1),r(1),s(2),a(2),r(2),\ldots ,s(T-1),a(T-1),r(T-1)\right\}$. In every episode, the ITE performs the action $a\left(t\right)$ to select the haptic codecs relating to the policy $\pi \left(a_{t}|s_{t}\right)=P\left[A=a_{t}|S=s_{t}\right]$. The agent selects the corresponding to the received observation from the environment and here we called it personalized learning action (PLA). After that agent receive a reward of $r\left(t\right)$ and transfer to the next state. The goal is to find the optimal policy value, $V^{\pi }\left(s\right)=E_{\pi }\left[\Sigma_{t=1}^{T-1}r\left(t\right)\right]$ for each state after actions select and maximize it using the optimal value function value $V^{\pi^{*}}\left(s\right)=\max_{\pi }V^{\pi }$. To achieve this, we utilize the Q-learning algorithm to solve the MDP problem and maximize the state value to select the optimal action.

In the literature, Q-learning and its variant algorithms have shown robust results to solve MDP problems. It is an off-policy approach that can be employed for any solution in the MDP framework. On the other hand, Q-learning uses, parametric operations to approximate the Q-function and defined it as follows:(3)$$Q\left(s_{t},a_{t}\right):=Q\left(s_{t},a_{t}\right)+\alpha \left(r_{t}+\gamma \times \max_{a}Q^{\prime }\left(s_{t+1},a\right)-Q\left(s_{t},a_{t}\right)\right),\;$$
where $Q(s_{t},a_{t})$ is the Q-value and is calculated in the Q-table. α∈(0<α≤1) is a learning rate coefficient, $r_{t}$ is the immediate reward agent receive after performing action $a_{t}$ on state $s_{t}$, and γ is the discount factor that helps to determine the importance of the future reward. $Q^{\prime }\left(s_{t+1},a\right)$ calculates the best Q-value form the next state. Figure 5 presents the high-level systematic and implementation flow of the proposed ITE framework. In the proposed ITE, firstly the system initialized the user interaction interface to interact with the system and parametric setting default values or as per user definition. It set the packet rate, link bandwidth, link latencies, packet size, and Internet protocol (IP) address. After initialization system predicts the network congestion and measures the QoS and QoE of the application requirement to the predicted state. If the condition meets, then the system continues with the same codecs if not then the ITE agent selects the optimal codecs corresponding to the state of the system. In the next section, we will present the experiment analysis to prove the effectiveness of the proposed system. The list of default parameter values and settings for experiments are summarized in Table 2.

## 4. Result and Discussion

In this section, we discuss and highlight the efficacy of the proposed ITE framework with haptic-enabled bilateral teleoperation system scenarios using different simulation parametric settings. Default parameter values and settings for experiments and the list of haptic codecs are presented in Table 1 and Table 2. The presented ITE is openly accessible at (https://github.com/zubair1811/IntelligentTactileEdge2022, accessed on 15 September 2022) for the reproducibility of our work and the ease of fellow researchers.

### 4.1. Simulation Setup

Our simulations environment is created in Python on a machine with an Intel Core i7 processor, 16GB memory, NVIDIA GeForce GT 1030 graphics card, and 64bit Linux (Ubuntu 18.04) operating system. The LSTM algorithm is programmed in the TensorFlow framework, and the Q-learning algorithm is programmed in Python. A Mininet emulator is utilized to design the communication network. Users may simulate actual network topologies with this tool. It also has built-in support for SDN architecture. For the experimentation, we utilized the publicly available 3 Degrees of Freedom (3DoF) teleoperation system haptic dataset [33]. A human operator at the master side uses a Phantom Omni haptic device and interacts with the virtual environment, which serves as a controlled domain, in order to capture the haptic traces. The position, velocity control signals of the human operator at the master side, and force feedback from the salve side for the 3DoF teleoperation dataset are depicted in Figure 6.

### 4.2. Latency Characterization

In haptic communication, QoS parameters are characterized by E2E delay, reliability, and synchronization [34]. The latency requirements for Tactile Internet applications depend on the type of application and dynamics of the remote environment. Some applications such as serious games (tele-soccer) demand latency value ≤10 ms and virtual reality-based teletherapy for phobia treatment E2E delay latency requirements (50 ms) with the packet failure rate $(10^{-3}$–$10^{-7})$. The latency and packet loss rate deeply impact QoE and the quality of task for haptic-driven applications. The work in [35] presents the role of the Tactile Internet in industrial applications. The authors explored the emerging tactile industrial services and compared their QoS requirements with conventional industrial applications. For example, remote control industrial applications including, process automation, monitoring, maintenance, and fault reporting demand cycle time (≤50 ms), data rate (1∼100 Mbps), latency (≤50 ms) with packet loss rate around ≤$10^{-7}$. Characterization of the latency for teleoperation systems is vital because it helps to provision QoS and QoE. Figure 7 depicts the analysis of the teleoperation system in terms of the E2E delay.

To investigate the effect of network congestion with no external host and with multiple external hosts during haptic communication among the master and the slave, a simple haptic-enabled teleoperation system is developed and simulated for the number of external hosts = 1 to 20. The latency investigation with direct haptic communication without the external host, with the number of hosts = 10, 15, 20 is illustrated in Figure 7b–d, respectively. The analysis clearly shows that the trend with the increasing number of hosts increases the round trip time that a packet experiences from an average of 3.4 to 1159 ms. In Figure 7a, the direct haptic communication without an external host between master and slave, the packet latency is between 1 to 3.4 ms as compared to Figure 7c,d, where the average latency values are between 23 to 1159 ms. To demonstrate this latency analysis in-depth, Figure 8a–d depicts the histogram of the packet latencies. This analysis reveals the E2E delays of the haptic data traffic due to network load that induced congestion. Similar to Figure 7a–d, the result in Figure 8a–d indicates that the increase in the number of external host network experience congestion and lead the system to unstable and degrade the application performance. Figure 8a, indicates that most of the latency of the packet lies under 2 to 3 ms, and the frequency of the number of the packet under 4 ms is more than 95%.

From Figure 8b–d, it can be clearly seen the latencies of the haptic traffic increase. In Figure 8b, most of the latency of the haptic packets centered between 20 to 60 ms, for Figure 8c it is between 600–1000 ms with an average of 849 ms and Figure 8d it is centered around 800 to 1500 ms with an average of 1159 ms. The packet interarrival times of teleoperation traces are also investigated and illustrated in Figure 9. The control signals as defined in the 3DoF teleoperation dataset are transmitted as a packet to the slave and the slave backward the haptic force feedback in response. The packet interarrival times are calculated as follows:(4)$$I_{i}^{\;c}=T_{i}^{\;c}-T_{i-1}^{\;c},$$
where *c* refers to the control signals, *i* is the sample that is transmitted, and *T* indicates the time instantly. In simple, it can be written the interarrival time of the packet $I_{i}^{\;c}$ is the difference between the current sample *i* at the time instant $T_{i}^{\;c}$ and the previously transmitted signals at $T_{i-1}^{\;c}$. Figure 9a, depicts the packet interarrival time for control signals and Figure 9b illustrates for the force feedback signals.

### 4.3. Learning Convergence

This section presents the study of the Temporal Difference (TD) error made by the proposed system in the learning process over time to explore the different parametric settings to get faster convergence of the system. The main goal is to attain a better convergence with a smaller number of training epochs to reduce the computation overhead. The TD is defined as follows:(5)$$\Delta Q=TD\left(s,a\right)=r_{t}+\gamma \times \max_{a}Q^{\prime }\left(s_{t+1},a\right)-Q^{o}\left(s_{t},a_{t}\right),$$
where $r_{t}+\gamma \times \max_{a}Q^{\prime }\left(s_{t+1},a\right)$ show TD for newly computed Q-value, while $Q(s_{t},a_{t})$ refers the previous Q-value for the state $s_{t}$. The ΔQ refers to the change in the Q-value and is obtained by subtracting the prior value from the calculated target value.

To study the effect of the learning rate α, on the TD error the α values from 0.1 to 0.5 is adopted. Practically, the values of the learning rate coefficient are selected in the range of 0.1 to 0.9, where α value 0 implies no learning of the agent and α = 1 mean the agent just focused on current knowledge and ignored the previous knowledge. In the proposed ITE we focus on the previous and current level of congestion in the network, so we investigate the ITE on learning rate values α=0.1∼0.5. The γ highlights the importance of the future reward, and the value of discounted factor γ = 0.8 is adopted for experimentation. The system explored the hidden pattern to recommend the optimal action within fine time episodes (T=40). The analysis of the TD error values for the ITE framework is illustrated in Figure 10 and Figure 11 with values of α 0.1 and 0.5, respectively. Figure 10a–d depicts the TD for the number of training epochs = 1, 2, 5 for the number of external host = 5, 10, 15, 20 during haptic communication between master and slave. Figure 10a with 5 external host depicts more TD error than Figure 10c,d, where the value of TD error lies between 0 to 0.05. Similarly, in Figure 10a, the ITE system with a value of learning rate 0.1 and epochs = 2–5 shows less error. However, in Figure 10d, with external host = 20 the shows convergence with number of training epoch = 1. The ITE is also simulated with a higher value of α = 0.5 to explain the better convergence as depicted in Figure 11. Figure 11a–d clearly sgows that the system with α = 0.5 reveals a significant decrease in TD error. As compared to Figure 10a with Figure 11a from episodes = 1–10 show less error with epochs = 2 and 5. Similarly, Figure 11c for 15 hosts with epochs 20 to 40 shows fewer errors than Figure 10c. The experiment observation denotes that, with learning rate value α = 0.5 and the number of epochs = 2–5, the proposed ITE shows less error than other parameters settings and shows fast convergence. So, this proves our attention to adopt the α = 0.5, γ = 0.8 and 2–5 number of training epochs. Moreover, show the better convergence of the proposed system with the selected parametric values throughput, RTT, and power function graphs are also presented.

The E2E performance of RTT during haptic communication with the external host are shown in Figure 12 and Figure 13. Figure 12 illustrates the simulation results of the ITE in terms of RTT, whereas Figure 12d with 20 external host shows that the proposed system focus to endure the QoS/QoE of the tactile application as compared RTT with our system with 20 hosts in Figure 7d. The Flow Queue Controlling Queue Delay (FQ-CoDel) scheme without any intelligence was used as a baseline model to illustrate the effectiveness of the proposed ITE. The RTT simulation results of the FQ-CoDel with the random strategy to select haptic codecs are shown in Figure 13. Figure 14 also shows that the average throughput of the system fluctuates around 1.5 Mbps, which ensures the stability of the network and controls the congestion level while using the proposed ITE. Finally, a comparative analysis of the power function for ITE and FQ-CoDel-Random is depicted in Figure 15, which shows the cumulative power of the connection during haptic communication with the presence of all external hosts. The results indicates that the ITE outperforms the FQ-CoDel-Random model while providing QoS and QoS requirements of the emerging service of Tactile Internet. Since the proposed ITE simulation has one nested loop (Outer loop for iteration and inner loop for episodes), the time complexity of the ITE is $n\;\times \;n=O(n^{2})$.

## 5. Discussion and Future Work Directions

In [35], we proposed the network infrastructure to simulate and investigate the delay-sensitive and loss-prone tactile industrial application. A simulator, known as IoTactileSim aims to provide a tool to examine the strict QoS and QoE requirement for tactile applications. Recent work [36] on QoS provisioning explored the haptic coding, control systems, and intelligent prediction techniques with a focus to deal with mission-critical and delay-sensitive stringent QoS and QoE requirements like ultralow latency, ultrahigh reliability, high availability, and ultrasecurity. This paper aims to propose an ITE framework to utilize the recently proposed haptic traffic reduction or compression methods to adjust the packet rate and packet size based on the level of congestion in the network and improve QoS and QoE. The proposed ITE observes network impairments during remote interaction and compares them with QoS/QoE specifications. The threshold-based approach is used to ensure the QoS and QoE requirements, and to perform the optimal action for selecting the best codec to control the threat of QoS violation. There should be an adaptive approach to defining the application-specific threshold corresponding to network resources. Moreover, haptic traffic communication between the master and the slave is performed over the TCP. However, some studies in the literature suggest that the user datagram protocol is preferable due to the low header size and processing cost. Therefore, we intend to implement such real-time application protocols in our future research to overcome the extra computational and communication cost. To demonstrate the fruitfulness of the proposed system, there is a need to integrate a real-world haptic device at the master to transmit the haptic data and perform the industrial remote task at the salve side. In this regard, in future work, a haptic device will be utilized instead of a haptic dataset to controls the real-world tactile enabled remote control industrial applications. It will help to understand the more practical challenges related to haptic-driven applications in real-world scenarios.

## 6. Conclusions

In this work, we proposed an Intelligent Tactile Edge framework to predict the congestion level at the network layer by utilizing the ECN mechanism and adjusting the packet size and packet rate of the haptic transmission at the application layer by employing different haptic codec schemes with a focus ensure QoS and QoE. The proposed ITE utilizes the LSTM model to estimate the network congestion and then the Q-learning algorithm applies to take the action and select optimal haptic codecs to solve the communication network issues (transmission delay, jitter, and packet loss). We employ the power function of the connection as a reward function to optimize the Q-learning algorithm and find the pattern. The proposed ITE was investigated on the Tactile Internet standard 3DoF teleoperation traffic dataset. The simulation results suggest that the ITE is able to ensure the QoS and QoE requirements of the haptic-enabled bilateral teleoperation application. The proposed framework was simulated on different parametric settings to tune the algorithm and show the algorithm convergence to select the optimal parametric configuration. In the recent future, we plan to implement the proposed system in real-work application with a haptic device and extended it with advanced machine learning techniques such as federated learning-aware approaches at the ITE to support emerging technologies.

## Figures and Tables

**Figure 1 sensors-22-08001-f001:**
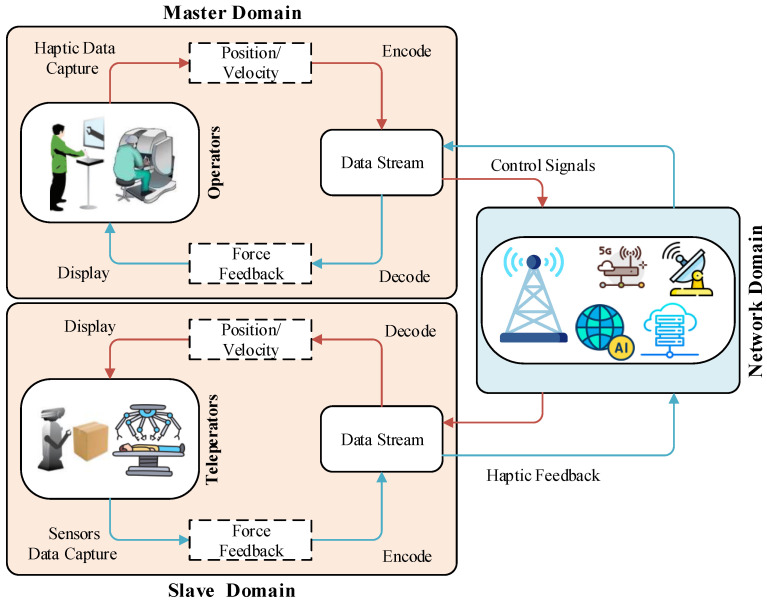
A high-level basic architecture design of the teleoperation system.

**Figure 2 sensors-22-08001-f002:**
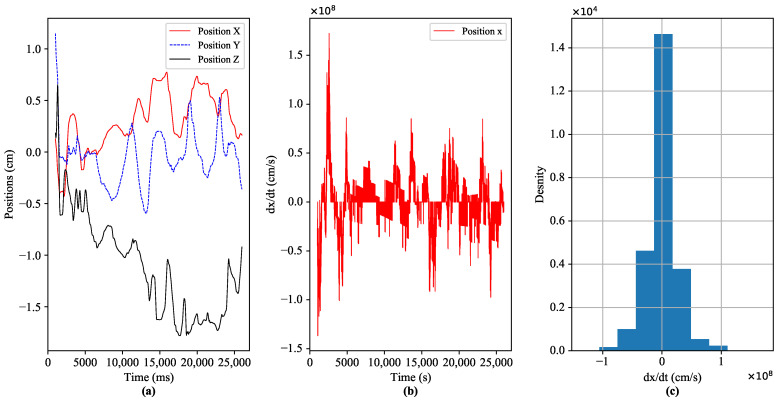
Dynamic interaction of the operator (master) with remote virtual application (slave) through a Falcon haptic device, (**a**) represents the positions of the human operator’s hand at master side, (**b**,**c**) illustrates the in-depth exploration of position at x-axis.

**Figure 3 sensors-22-08001-f003:**
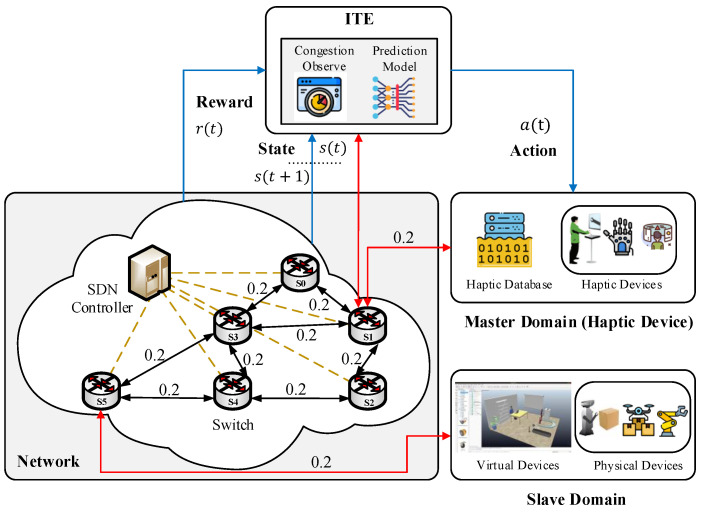
Indepth overview of the proposed ITS.

**Figure 4 sensors-22-08001-f004:**
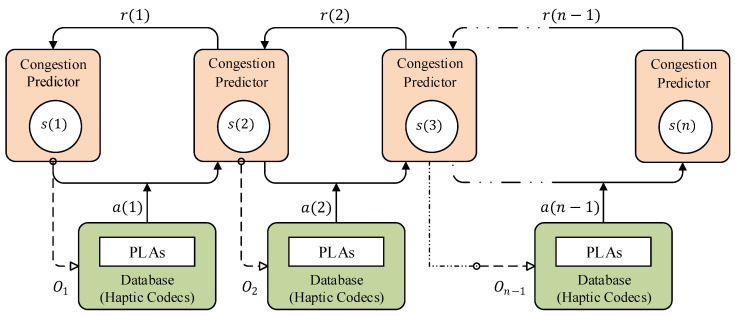
State transition model of the dynamic switching strategy: (1) prediction model estimates the state $s_{i}\left(t\right)$ of the network congestion by observing the ECN marked packets as $O_{i}$; (2) following the policy, the proposed scheme selects the haptic codecs from the database by action a(i); and (3) transit the system into the next state $s_{j}\left(t+1\right)$ by receiving reward r(i).

**Figure 5 sensors-22-08001-f005:**
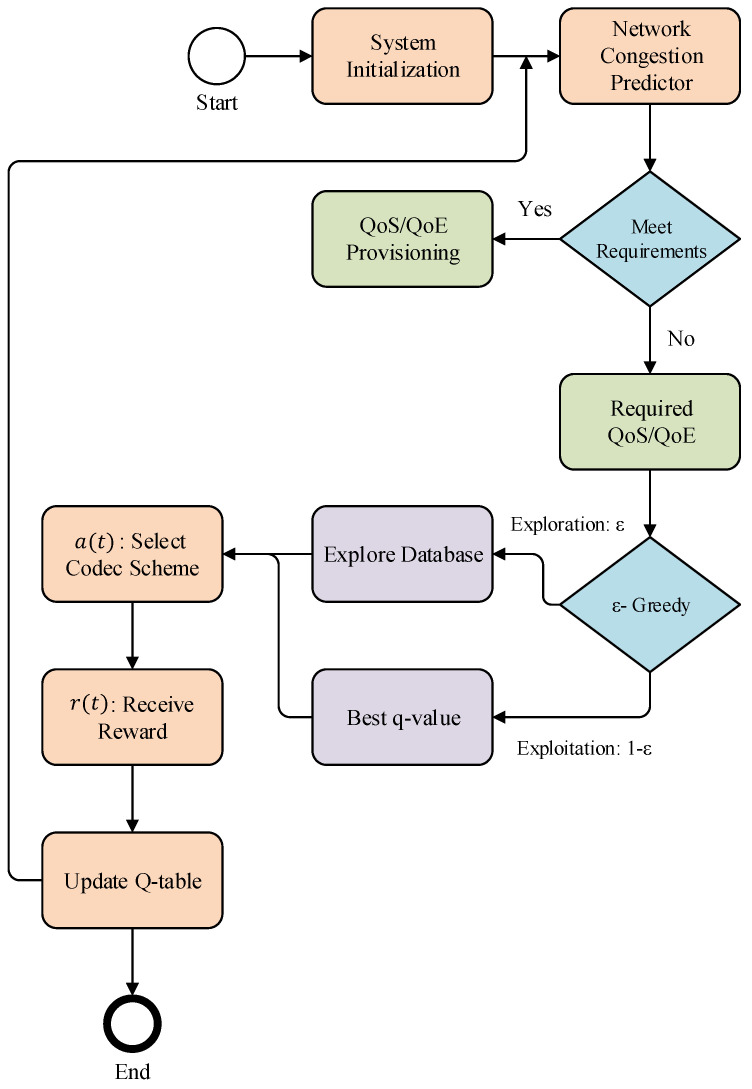
Flowchart shows the systematic flow of the proposed ITE framework.

**Figure 6 sensors-22-08001-f006:**
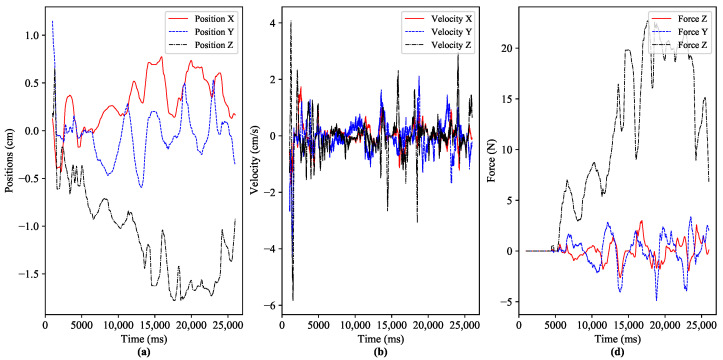
Dynamic interaction of the human operator with virtual application via haptic device. (**a**) positions of the human operator’s hand at master side device, (**b**) velocity traces of the operator (**c**) force data traces of the teleopertor *x*,*y* and *z*-axis.

**Figure 7 sensors-22-08001-f007:**
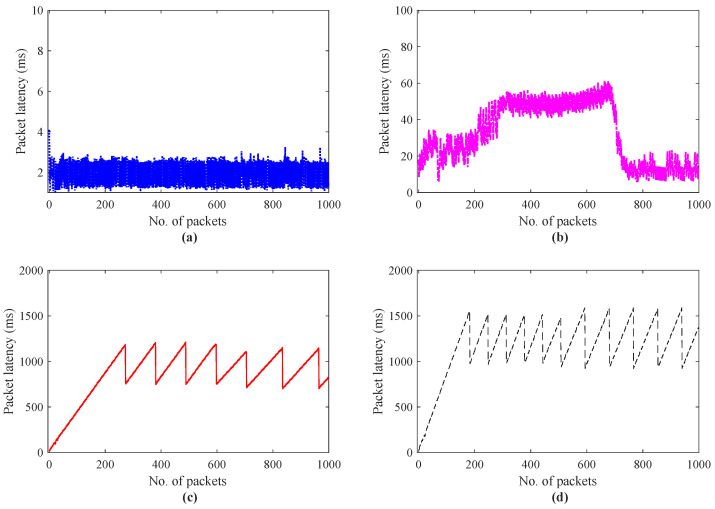
Packet latency investigations for 3Dof haptic data transmission; (**a**) master and slave direct communication (no external host are involved), (**b**) External Host = 10, (**c**) External Host = 15, (**d**) External Host = 20.

**Figure 8 sensors-22-08001-f008:**
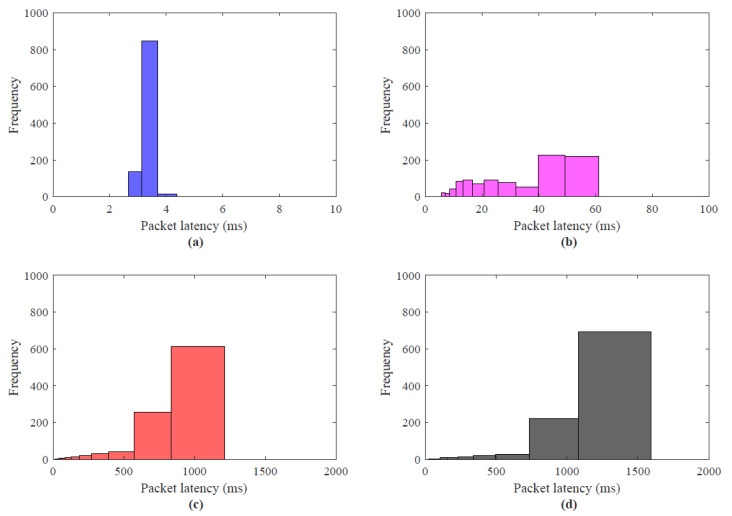
Packet latency histogram for 3Dof haptic data transmission; (**a**) master and slave direct communication (no external host are involved), (**b**) External Host = 10, (**c**) External Host = 15, (**d**) External Host = 20.

**Figure 9 sensors-22-08001-f009:**
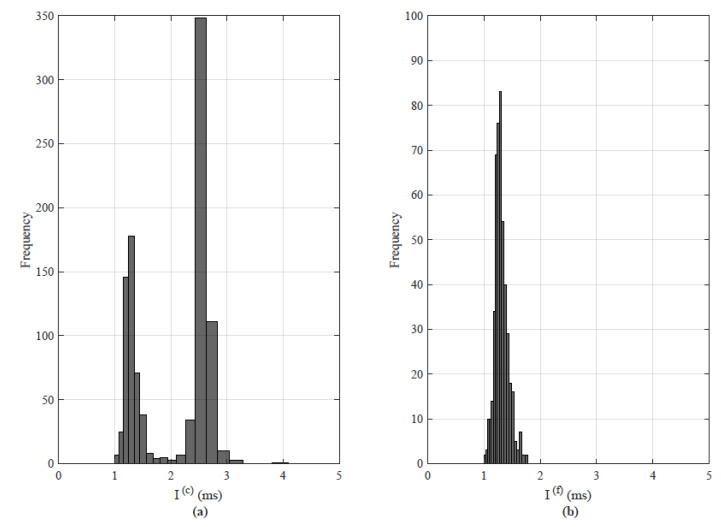
Interarriavl packet time for 3DoF haptic teleoperation traces: (**a**) command/control signals and (**b**) feedback signals.

**Figure 10 sensors-22-08001-f010:**
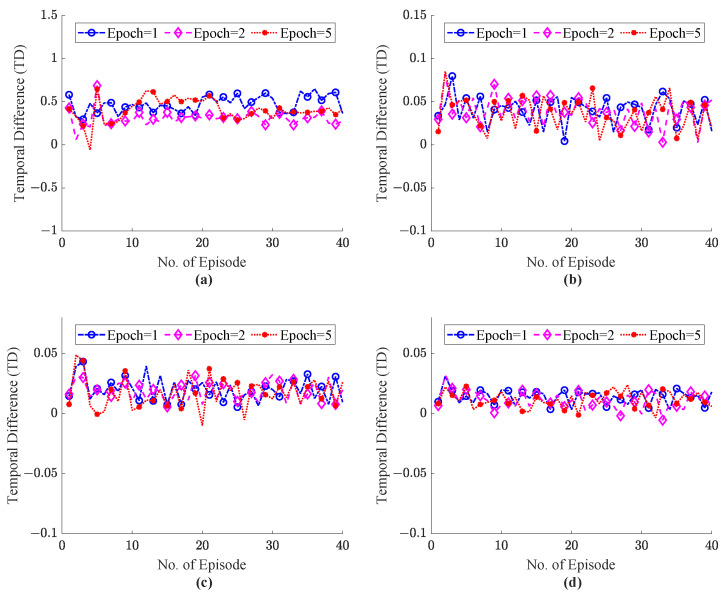
Temporal difference error comparison with parametric value α = 0.1 and the number of training epochs ranging 1–5; (**a**) External Host = 5, (**b**) External Host = 10, (**c**) External Host = 15, (**d**) External Host = 20.

**Figure 11 sensors-22-08001-f011:**
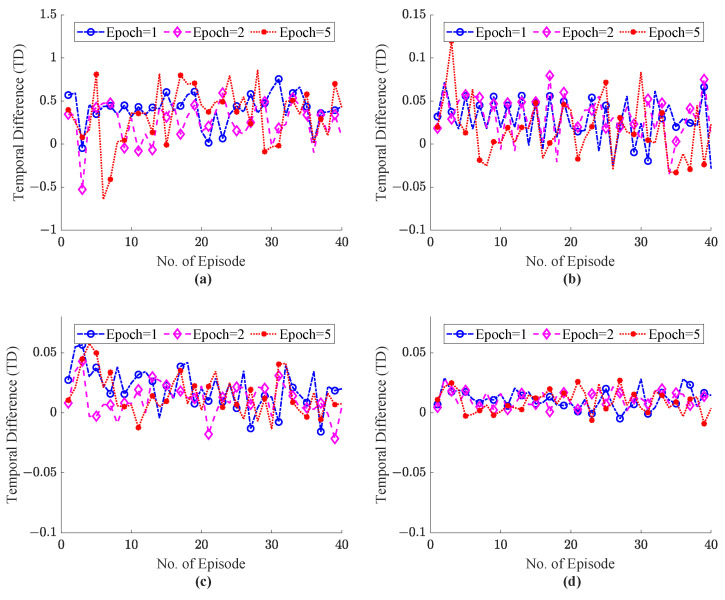
Temporal difference error comparison with parametric value α= 0.5 and the number of training epochs ranging 1–5; (**a**) External Host = 5, (**b**) External Host = 10, (**c**) External Host = 15, (**d**) External Host = 20.

**Figure 12 sensors-22-08001-f012:**
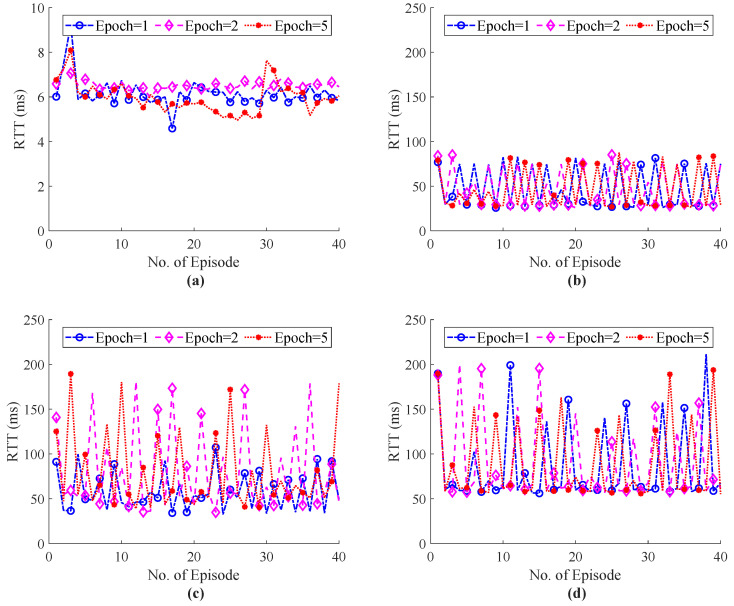
Total E2E RTT of the system with ITE during haptic communication with different number of training epochs and external hosts; (**a**) External Host = 5, (**b**) External Host = 10, (**c**) External Host = 15, (**d**) External Host = 20.

**Figure 13 sensors-22-08001-f013:**
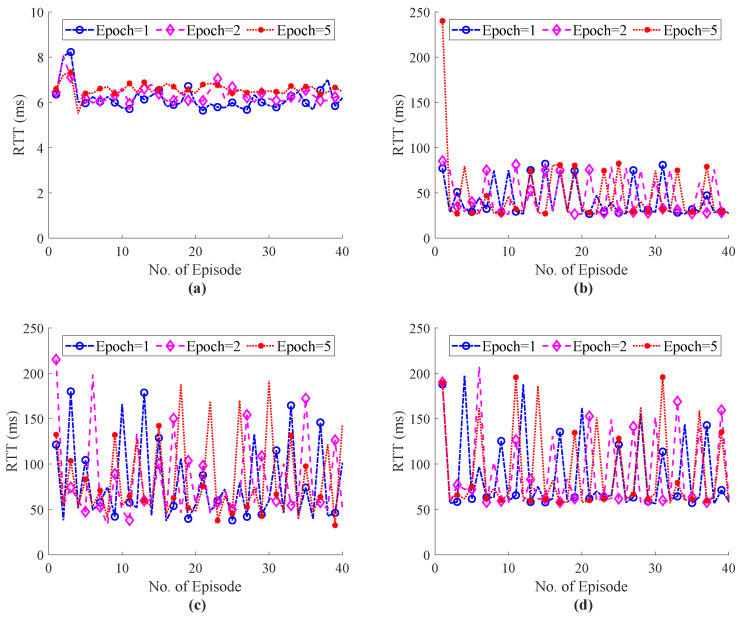
Total E2E RTT of the FQ-CoDel-Random (Baseline) during haptic communication with different number of training epochs and external hosts; (**a**) External Host = 5, (**b**) External Host = 10, (**c**) External Host = 15, (**d**) External Host = 20.

**Figure 14 sensors-22-08001-f014:**
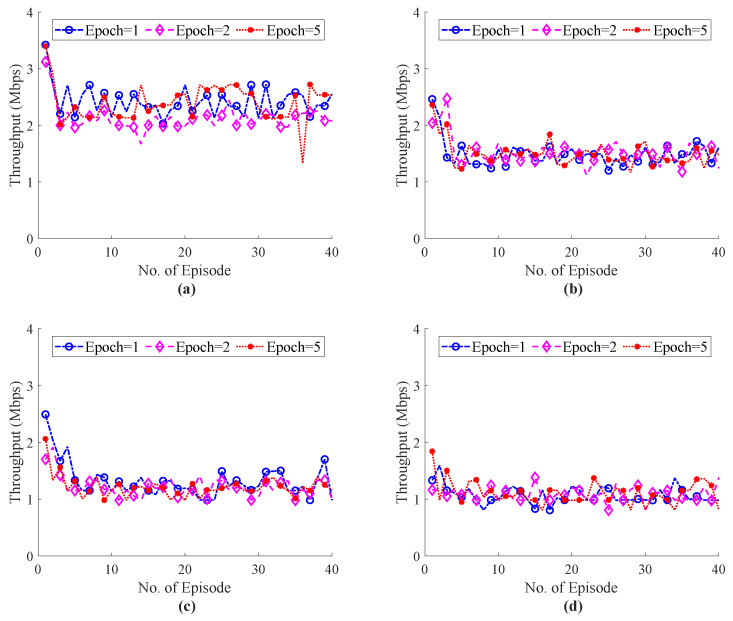
Total E2E throughput of the system during haptic communication with different number of training epochs and external hosts; (**a**) External Host = 5, (**b**) External Host = 10, (**c**) External Host = 15, (**d**) External Host = 20.

**Figure 15 sensors-22-08001-f015:**
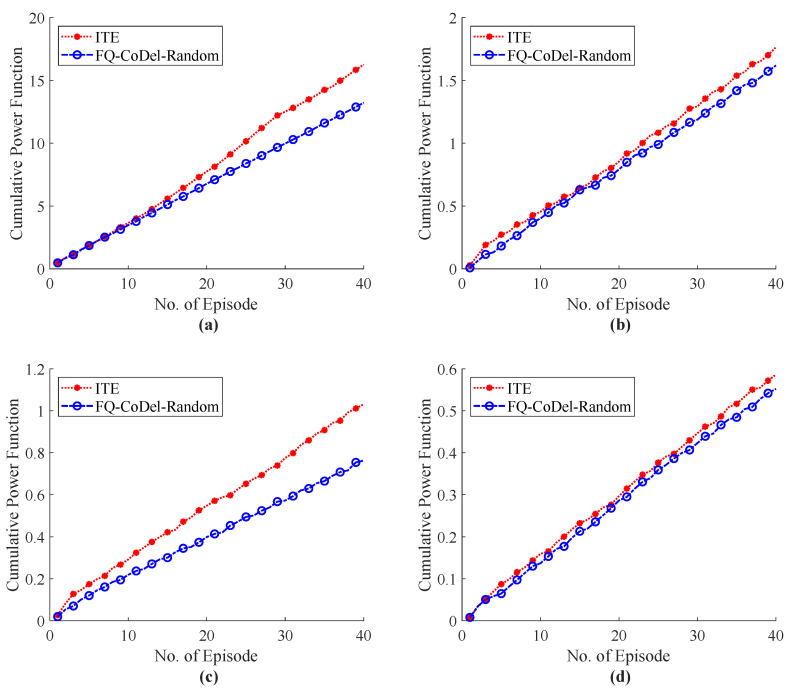
Commutative power (Reward=Throughput/RTT) of the connection determine during simulation experimentation for ITE and baseline model with different number of training epochs and external hosts; (**a**) External Host = 5, (**b**) External Host = 10, (**c**) External Host = 15, (**d**) External Host = 20.

**Table 1 sensors-22-08001-t001:** Summary of the haptic codec techniques for emerging Tactile-driven services. The Effectiveness column indicates the level of significance like Significant (Sig.), Not Significant (Not Sig.), and Mixed. The Sig. means schemes use standard database and baseline comparison, Not Sig. means use a standard database but lack baseline comparison and the Mixed means presented scheme shows baseline comparison in the absence standard database.

Proposed Technique	Improved QoS/	Effectiveness	Key Contribution
	QoE Factor(s)		
OAHS [21]	SNR	Mixed	Propose an adaptive sampling method to tune PDb thresholding in real time with a focus on minimizing network impairments for the telehaptic system.
Data-drive [22]	SSIM	Sig.	To meet the stringent requirements of Tactile Internet ultralow delayed, the data-driven technique was proposed.
PLC [23]	SNR, HSSIM	Sig.	To minimize the coding delay, and coding loss and to improve the bit error rate the E2E perceptual-lossless codec was presented to ensure the quality of tactile services.
PVC-SLP [24]	SNR, PSNR	Sig.	A sparse linear prediction coding-based vibrotactile technique was presented, where a cutaneous sensitivity function was employed to improve the quality of vibrotactile signals.
VC-PWQ [25]	SNR, PSNR	Sig.	Introduce a hybrid scheme by coupling wavelet transformation and vibrotactile perceptual model for haptic-driven applications.
Peak-Suppressing [26]	MSE	Mixed	Present a peak-suppressing scheme to adjust the PDb and reduce packet rate with a focus to minimize the network load that leads to delay and packet loss.
Adaptive Sampler [27]	MOS	Not Sig.	An adaptive sampling scheme for selected haptic sample transmission was presented to guarantee the QoS of the teleoperation systems.

**Table 2 sensors-22-08001-t002:** List of parameters and settings used in experimentation.

Parameters	Settings Used
**Simulation Setup**	
Operation system	Ubuntu 18.04
Programming	Python 3.8
Network design emulator	Mininet 3.6.5
**Network Emulator**	
Network topology	Mesh network of switches
IP suite	Transmission control protocol
Software switch type	Open vSwitch 2.9.8
SDN based controller	OVS-controller
SDN controller protocol	OpenFlow
Link delay	Shortest route 1.6 ms
Link bandwidth	20∼100 Mbps
Packet sampling Rate	1 kHz∼3 Mbps
**Q-Learning Algorithm**	
Learning rate (α)	0.1, 0.5
Discounted factor (γ)	0.8
Exploitation/Exploration coefficient (ε)	0.5
Episodes (T)	40
No. of Training Epochs	1–5
Set of action space (A)	7
Set of state space (S)	100

## Data Availability

The 3DoF haptic datasets used in this work are available online with open access for academic research use. The static and dynamic interaction haptic dataset is available online at https://cloud.lmt.ei.tum.de/s/4FmHUCsoUvwRle3 (accessed on 15 September 2021), and the simulation experiment results files are availabe at https://github.com/zubair1811/IntelligentTactileEdge2022, accessed on 15 September 2022).

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
