# Peer review of "Reinforcement Learning-Aided Edge Intelligence Framework for Delay-Sensitive Industrial Applications"

_sensors, 2022, doi:10.3390/s22208001_

Round 1

Reviewer 1 Report

- Background and related work about software-defined networking (SDN) are missing (in the introduction and related work). SDN is mentioned in Fig. 3, but it is not explained. How is it related to Tactile Internet?

- Please explain why the algorithms or models (LSTM, Q-learning) were selected. There are more advanced algorithms and models for predictions and learning, e.g., Double Q-learning, Deep Deterministic Policy Gradient.

- The experiment is missing the comparison with other methods or algorithms. This is important to show the advantages of the used algorithms or models.

- Some Abbreviations are used without full explanation in the text or in the Abbreviations table, e.g., in Table 1, SNR, SIG, etc. 

- Some content is similar or exact to the published publications (checked by Turnitin. Without the references, the similar rate is 25% - which is high).

- References: The format is not consistent and the references are missing some information, e.g., volume. Please check all references. For example, [29] and [30] have the same authors but their names are different. [30] is missing the volume. [2] Ieee?

- Presentation:

+ Table 1, In the Effectiveness column, Sig -> SIG, Not Sig -> NOT SIG, mixed -> Mixed.

+ After an equation, the "where" should not have a tab space. After an equation, some have a comma, some have not.

+ In Figures, the text is small, which is not easy to read. The Figures are also small in the experiment section.

+ The hyperlink is broken, e.g., (http://mininet.org/,accessedon15September2021), in which accessedon15September2021 has no space between the words, and when clicking the link, it cannot open.

+ Please revise the writing also. For example, after a dot, it has to have a space (see the caption of Table 1).

Reviewer 2 Report

- The abstract can include numerical results obtained.

- There have been many related works. What are the gaps identified that motivated the current research?

- Some of the recent works such as the following can be discussed A survey on Zero touch network and Service (ZSM) Management for 5G and beyond networks, AI and 6G Security: Opportunities and Challenges, 

- Proposed framework section should include a discussion on the novelty of the proposed RNN based approach.

- COmpare the results obtained with recent state of the art.

- What is the computational and communication cost of the proposed approach?

- What are the threats to validity of the proposed approach?

Round 2

Reviewer 1 Report

The authors have addressed my comments thoroughly.

Reviewer 2 Report

The authors have addressed all the comments. I have no further comments/queries.